# Thermal Deformation Behavior and Interface Microstructure Analysis of 2205/Q345 Hot Compression Composite

**Xiaoyang Wang [1], Pengtao Liu [1,\*], Guanghui Zhao [1,2], Juan Li [1,2]** 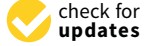 **and Chenchen Zhi [1]**

[1] School of Mechanical Engieering, Taiyuan University of Science and Technology, Taiyuan 030024, China;
201912180228@stu.tyust.edu.cn (X.W.); 2017012@tyust.edu.cn (G.Z.); 2021019@tyust.edu.cn (J.L.);
2020045@tyust.edu.cn (C.Z.)

[2] Shanxi Provincial Key Laboratory of Metallurgical Device Design Theory and Technology, Taiyuan University of Science and Technology, Taiyuan 030024, China

\* Correspondence: 2020120@tyust.edu.cn

**Abstract:** The high-temperature thermal deformation behavior of a 2205/Q345 specimen at 850–1100 °C and strain rate of 0.01–10 s$^{-1}$ was systematically studied by the Gleeble-3800 thermal simulator, which provided a theoretical basis for the optimization of a 2205 duplex stainless steel composite plate. It is found that the deformation resistance of 2205 steel was different from that of Q345 steel. Therefore, the Q345 steel deformed first, the degree of deformation was large, the degree of recrystallization occurred, and the grain was isometric and relatively large. The 2205 steel was subsequently deformed, the degree of deformation was relatively small, and the microstructure retained the original rolled and elongated structure. In particular, 2205 and Q345 show a coordinated deformation trend as a whole at 1050 °C and 1–10 s$^{-1}$. Under the action of shear stress, there are many fine grains at the composite interface.

**Keywords:** 2205 dual-phase stainless steel; hot compression composite; microstructure; hot deformation behavior

## 1. Introduction

Duplex stainless steel has more advantages than austenitic and ferritic stainless steel because of its good toughness, high strength, and corrosion resistance. Q345 steel is the most widely used light-alloy steel. Considering the economic cost and the properties of steel plates, the demand for duplex stainless steel-clad plates is increasing. It is widely used in petrochemical, shipbuilding, papermaking, nuclear energy, power generation, and other fields [1–5].

In duplex stainless steel, austenite and ferrite have great differences in composition, microstructure, and mechanical properties. During hot deformation, dislocation bunching and dynamic recovery are easy to occur in ferrite. However, the lower stacking fault energy of the austenite phase makes dislocation entanglement more easily, which results in the inhibition of dynamic recovery and other processes [6–9]. Moreover, the difference in mechanical properties and deformation behavior of the two phases makes the plastic deformation behavior more complex, which makes the hot working of the alloy more difficult [10]. In recent years, much attention has been paid to the hot deformation behavior of duplex stainless steel [11–14]. Song Yaohui analyzed the microstructure of the samples by electron backscatter diffraction (EBSD). The results showed that the content of the austenite phase decreases slowly with the increase of the strain rate, while the content of the ferrite phase increases [15]. Chen Lei et al. utilized thermal simulation experiments to study the thermal deformation behavior of 2205 duplex stainless steel and analyzed the corresponding microstructure evolution law [16]. The results showed that austenite is distributed in the ferrite matrix, and the volume fraction of ferrite increases with increasing temperature. However, there was little research on the hot deformation behavior of the 2205 dual-phase

steel composite plate. Liu Yanping analyzed the experimental research on the performance of the 2205/Q345R explosive composite board, and he focused on the analysis of the effect of the heat treatment process on the structure and properties [17]. Explosive compounding technology was adopted by Bi Zongyue et al. to explosively compound 2205 duplex stainless steel and carbon steel Q235 for the preparation of large-area duplex stainless steel clad plates that achieved metallurgical bonding [18]. A. Momeni analyzed the hot deformation characteristics of 2205 duplex stainless steel by using constitutive equation and machining diagram and found that the dynamic recovery mechanism can effectively prevent the occurrence of flow instability at low and medium strain rates [19].

Based on the above analysis, in order to explore the thermal cladding process of 2205 duplex stainless steel/Q345 carbon steel, this paper adopted the thermal simulation method to systematically study the high-temperature thermal deformation behavior of 2205 duplex stainless steel and Q345 carbon steel that was carried out under different temperatures and strain rates. We studied 2205/Q345's thermal compression, composite thermal deformation behavior, and interface microstructure evolution law. It is expected to provide a theoretical basis for the optimization of the composite process of the 2205 duplex stainless steel composite plate.

## 2. Experimental Materials and Methods

### 2.1. Materials

The 2205 duplex stainless steel and Q345 carbon steel used in this paper are from 16 mm thick steel plates, and their chemical compositions are displayed in Table 1. The initial microstructures of 2205 duplex stainless steel and Q345 carbon steel are shown in Figure 1. The Q345 carbon steel was composed of uniform and fine grains with sizes of 10 μm. The grains of 2205 duplex stainless steel are elongated with a width of 30 μm.

**Table 1.** Chemical composition of experimental steel (mass fraction, %).

| Steel | C | Cr | Mo | Ni | Si | S | Mn | P | N | Fe |
|-------|------|-------|------|-------|-------|-------|------|-------|--------|------|
| 2205 | 0.025 | 21.83 | 3.09 | 5.45 | 0.58 | 0.003 | 1.13 | 0.024 | 0.1699 | Bal. |
| Q345 | 0.20 | 0.25 | 0.10 | 0.012 | 0.50 | 0.035 | 1.70 | 0.035 | 0.012 | Bal. |

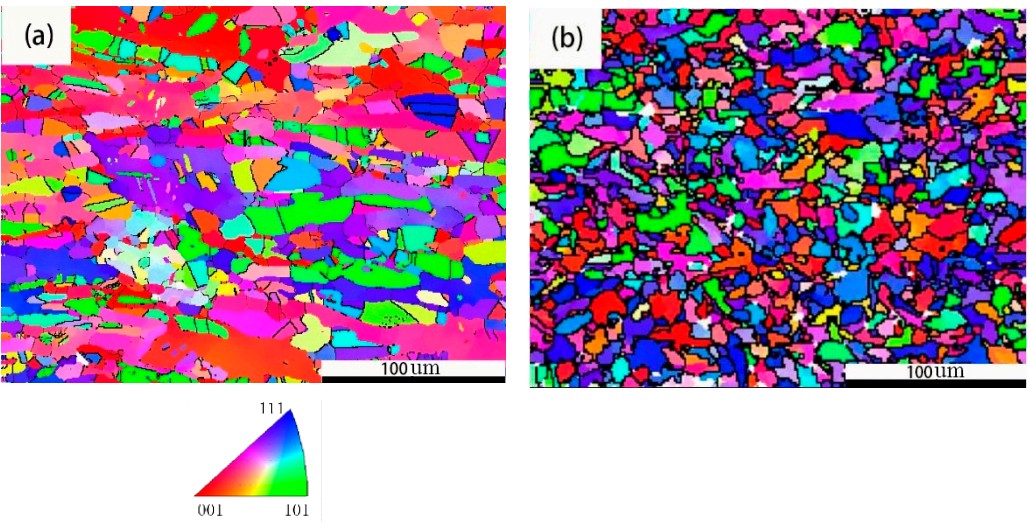

**Figure 1.** The initial microstructures of 2205 duplex stainless steel (**a**) and Q345 carbon steel (**b**).

### 2.2. Experiment Method

The experiment was carried out on a Gleeble-3800 thermal simulator. The sample was a cylinder with a diameter of 10 mm and a height of 15 mm. The end face of the cylinder was ground by a surface grinder to ensure the flatness of the end face so that there was no

gap between the two cylinder ends when they contact. When clamping the sample, graphite lubricating oil and tantalum pad was applied between the end face and the indenter, so as to reduce the friction between the indenter and the specimen, as shown in Figure 2. Before the experiment, two K-type stainless steel were welded to 2205 duplex stainless steel near the intermediate interface. After clamping, the sample chamber was vacuumed to make the vacuum degree lower than $1.0 \times 10^{-1}$ Pa, and then the operation program was started. The sample was heated to 1230 °C at a heating rate of 5 °C/s for 2 min to homogenize the structure. Next, at a temperature drop rate of 5 °C/s, the sample was reduced to the corresponding deformation temperature, after which undergoes thermal compression composite deformation. The deformation temperature (T) was 850–1100 °C, with an interval of 50 °C. The deformation temperature compression rates were 0.01 s$^{-1}$, 0.1 s$^{-1}$, 1 s$^{-1}$, and 10 s$^{-1}$. Meanwhile, the true strain was 0.92. After the thermal compression composite was completed, it was water-cooled to retain the thermally deformed structure.

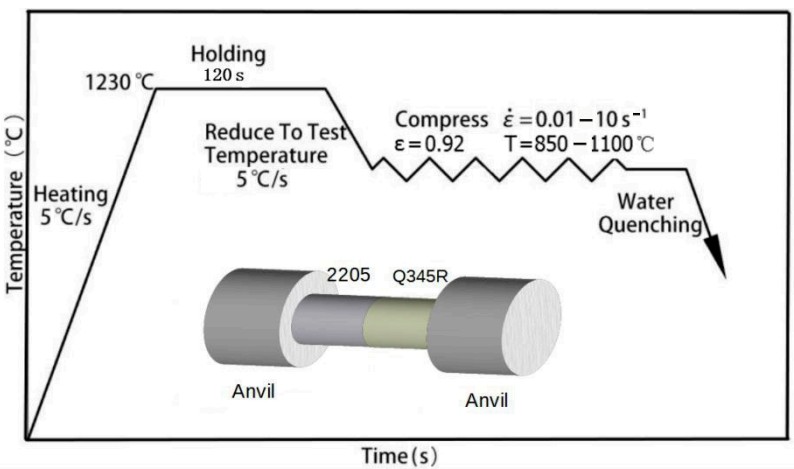

**Figure 2.** Flow chart of thermal compression composite.

After high-temperature compression, the sample was cut along the compression axis with wire cutting. The metallographic sample was polished by sandpaper, mechanically polished, and electrolytically corroded by a 20% NaOH solution, voltage 2.3 V, electrolysis time 100 s to complete the optical structure preparation. The cut section was applied for the preparation of the electron backscatter diffraction (EBSD) study sample. The sample was polished under standard metallographic procedures, and then a Buehler vibration machine was used for 2 h to ensure that any residual surface deformation was removed. Finally, a scanning electron microscope (ZIESS SIGMA FE-SEM) equipped with the Oxford EBSD probe was utilized to observe and analyze the microstructure morphology of the central area of the bonding interface of the composite board. The post-processing software Channel 5 was used for data analysis.

### 3. Results and Discussion

Figure 3 illustrates the macroscopic sample after 2205/Q345 thermal compression composite. From the analysis of different temperatures, it can be seen that the deformation behavior of the two samples at low temperatures (850–950 °C) is quite different. The deformation of Q345 is obviously greater than 2205. After reaching 1000 °C, the two materials increase with the temperature, and the deformation behavior tends to be similar. From the analysis of different deformation rates, it is obvious that as the deformation rate increases, the deformation of the material becomes more and more irregular, which is because the material does not have time to uniformly deform as the rate increases. At the same time, from Figure 3, the analysis demonstrates that when the deformation temperature is 1050 °C and the deformation rate is 1–10 s$^{-1}$, the deformation behavior of 2205/Q345 is almost similar.

| | 850 °C | 900 °C | 950 °C | 1000 °C | 1050 °C | 1100 °C |
|---|---|---|---|---|---|---|
| 0.01 s⁻¹ | 2205 / Q345 | | | | | |
| 0.1 s⁻¹ | | | | | | |
| 1 s⁻¹ | | | | | | |
| 10 s⁻¹ | | | | | | |

**Figure 3.** 2205/Q345 thermal compression composite macro sample.

### 3.1. Stress–Strain Curve

Figure 4 reveals the true stress–strain curve of the 2205/Q345 thermal compression composite test at the temperature, ranging from 850 °C to 1100 °C, and the strain rate of $0.01 \text{ s}^{-1}$–$10 \text{ s}^{-1}$. From the 2205/Q345 thermal compression composite stress–strain curve, it is found that in the initial stage of compression composite deformation, the flow stress increases rapidly with the increase of strain, after which the flow stress starts to increase slowly until it reaches the peak. Subsequently, the flow stress tends to decrease or maintain a steady state. The deformation curves are in the form of dynamic recrystallization.

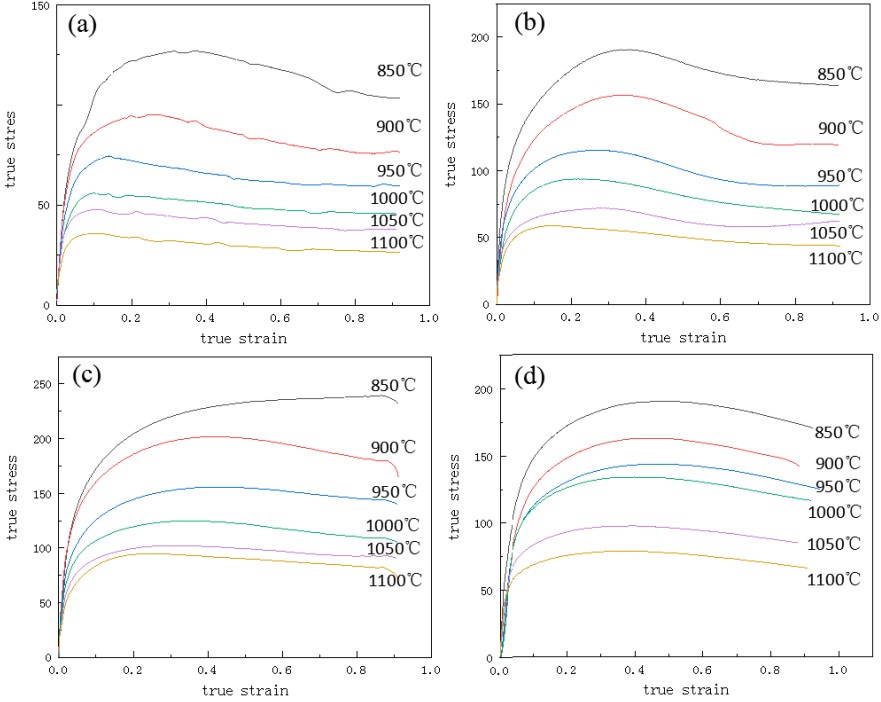

**Figure 4.** Flow stress curves under different conditions: (**a**) $0.01 \text{ s}^{-1}$, (**b**) $0.1 \text{ s}^{-1}$, (**c**) $1 \text{ s}^{-1}$, (**d**) $10 \text{ s}^{-1}$.

### 3.2. Microstructure Analysis of Composite Interface

Figure 5 shows the 500 times OM microstructure of the large deformation zone interface of the 2205/Q345 thermal compression composite with different temperatures and deformation rates.

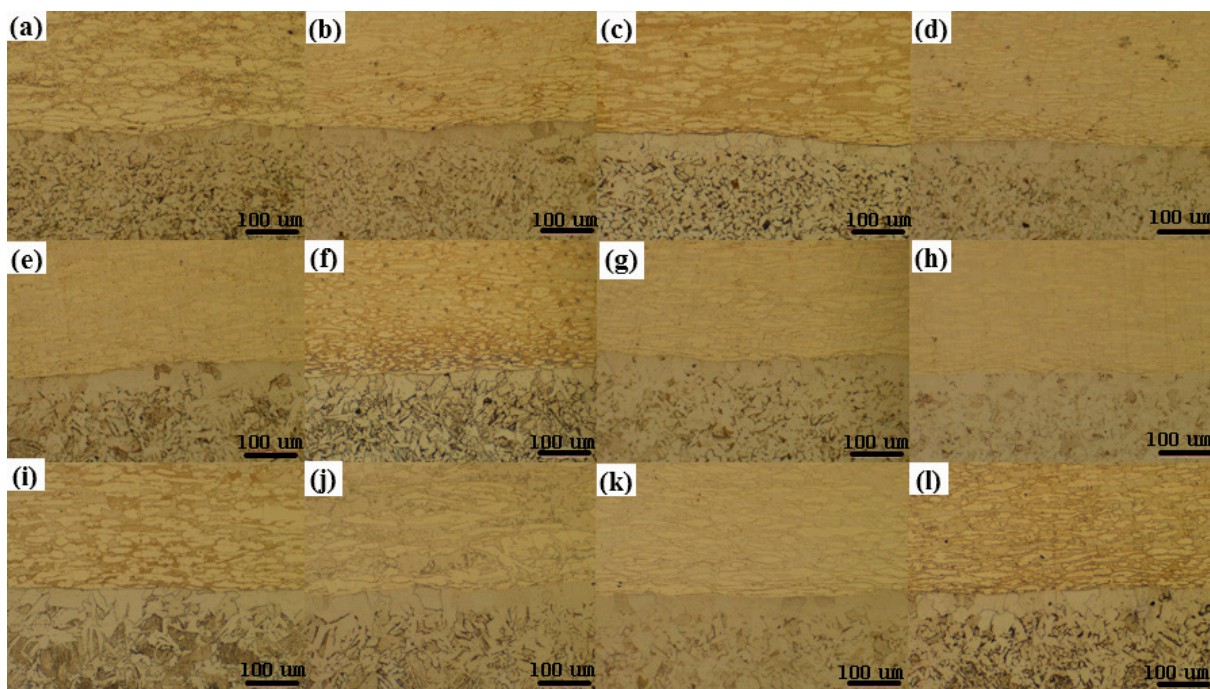

**Figure 5.** Microstructure of composite interface in large deformation area by optical microscope. (**a**) 850 °C, 0.01 s$^{-1}$; (**b**) 850 °C, 0.1 s$^{-1}$; (**c**) 850 °C, 1 s$^{-1}$; (**d**) 850 °C, 10 s$^{-1}$; (**e**) 1000 °C, 0.01 s$^{-1}$; (**f**) 1000 °C, 0.1 s$^{-1}$; (**g**) 1000 °C, 1 s$^{-1}$; (**h**) 1000 °C, 10 s$^{-1}$; (**i**) 1050 °C, 0.01 s$^{-1}$; (**j**) 1050 °C, 0.1 s$^{-1}$; (**k**) 1050 °C, 1 s$^{-1}$; (**l**) 1050 °C, 10 s$^{-1}$.

The upper layer is 2205 duplex stainless steel, and the lower layer is Q345 carbon steel. Between them, there is a bonding interface. Because the lower carbon steel is easy to corrode, its grain boundaries are clear and the structure is evenly distributed. Figure 6 shows the EBSD results of the 2205/Q345 thermal compression composite sample in order to further observe the structure on both sides. It is observed that Q345 equiaxed grains are coarse, and 2205 duplex stainless steel has fine grains and straight grain boundaries. The interface between them is straight without obvious gaps. During the experiment, there exists a difference in deformation resistance between 2205 dual-phase stainless steel and Q345 carbon steel that has a small deformation resistance. Therefore, in the process of deformation together, the carbon steel deforms first, with a large degree of deformation, recrystallization, and driving force. The crystal grains are equiaxed and relatively large. The 2205 dual-phase stainless steel is subsequently deformed, and the degree of deformation was relatively small. The microstructure retained the original rolled and elongated structure. At the composite interface, under the action of shear stress, there are many fine grains.

Figure 6 exhibits the change trend of 2205 and Q345 grains with increasing temperatures under the same strain rate. From Figure 6a,e,i, the grains of Q345 and 2205 grow with the increase of temperature when the temperature increases from 850 °C to 1000 °C and the strain rate is 0.01 s$^{-1}$. The trend is obvious because the high temperature and low strain rate compression leave the grains enough time to grow. When the strain rate is 0.1 s$^{-1}$, the growth trend of Q345 and 2205 grains becomes more obvious with the increase of temperature, and the obvious growth trend appears at the same time from 1000 °C to 1050 °C. When the strain rate is 1 s$^{-1}$, the Q345 grain growth trend is the most obvious between 850 °C and 1000 °C, while the 2205 grain growth trend is relatively small. At

this time, the two grains display a coordinated deformation trend. When the strain rate is 10 s$^{-1}$, the grain size of Q345 does not change remarkably as the temperature rises, while the grain size is smaller than that at low strain rates. The 2205 grains also retain the original rolled and elongated structure, which is most serious at 850 °C. Based on the above considerations, the strain rate 1–10 s$^{-1}$ is selected and the composite material has comprehensive excellent performance.

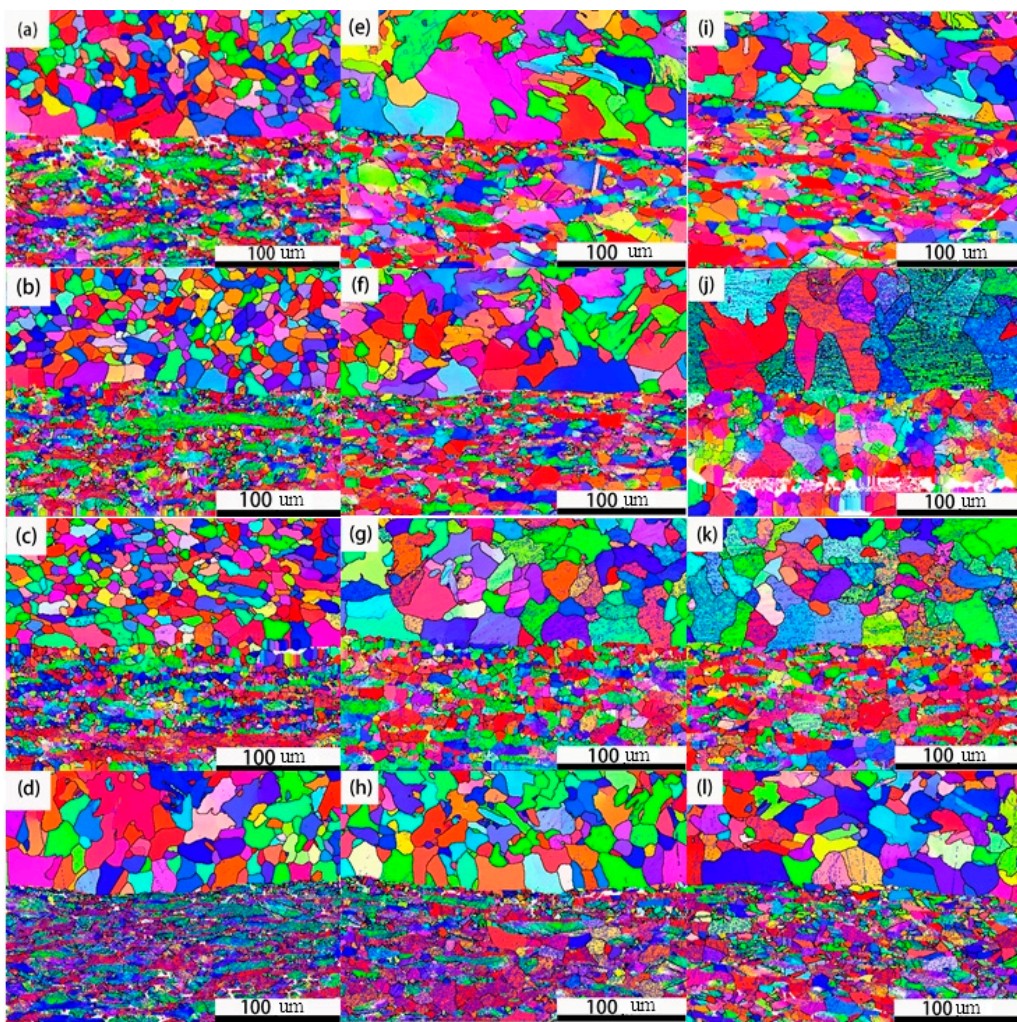

**Figure 6.** Composite interface IPF for 1000 times large deformation area (**a**) 850 °C, 0.01 s$^{-1}$; (**b**) 850 °C, 0.1 s$^{-1}$; (**c**) 850 °C, 1 s$^{-1}$; (**d**) 850 °C, 10 s$^{-1}$; (**e**) 1000 °C, 0.01 s$^{-1}$; (**f**) 1000 °C, 0.1 s$^{-1}$; (**g**) 1000 °C, 1 s$^{-1}$; (**h**) 1000 °C, 10 s$^{-1}$; (**i**) 1050 °C, 0.01 s$^{-1}$; (**j**) 1050 °C, 0.1 s$^{-1}$; (**k**) 1050 °C, 1 s$^{-1}$; (**l**) 1050 °C, 10 s$^{-1}$.

At the same time, from Figure 6, it can be intuitively discovered that the change rule of 2205 and Q345 grains are at the same temperature with the increase of strain rate. First of all, in terms of Figure 5a–d, the Q345 grain refinement degree is obvious but the size does not change significantly at a low temperature of 850 °C, with the increase of the deformation rate, and in the range of 0.01–1 s$^{-1}$. The Q345 grains grow significantly at 10 s$^{-1}$. For 2205, the grains are continuously compressed with the increase of strain rate, suggesting a strip shape. The grains of 2205 and Q345 are relatively large at 1000 °C and deformation rate of 0.01 s$^{-1}$. With the increase of the deformation rate, the deformation of 2205 grains continues to refine, whereas the grains of Q345 first become smaller and then change less, obviously. As the deformation rate increases from 0.01 to 0.1 s$^{-1}$ at 1050 °C, the grains of 2205 and Q345 both grow significantly, the original rolled strip structure of 2205 basically disappears, and recrystallization occurs during deformation that increases

with the deformation rate. The 2205 and Q345 grains have a coordinated deformation trend as a whole.

In polycrystals, the plastic deformation of a crystal grain cannot be independent, and it could inevitably cause coordinated deformation of other surrounding crystal grains. Similarly, in composite materials, the deformation of one component material can affect the coordinated deformation of another material at the interface. When deformed under external pressure, Q345 carbon steel and 2205 dual-phase stainless steel can coordinate deformation through the composite interface. It can be seen from the figure that there are various degrees of small grains at the composite interface. When the temperature is 1050 °C and the deformation rate is 1 s$^{-1}$, the Q345 carbon steel and 2205 dual-phase stainless steel on both sides of the composite interface indicate a relatively coordinated deformation trend.

Figure 7 reveals the different morphological characteristics of the phase and grain boundaries at the interface of the 2205/Q345 thermal compression composite sample under different strain rates at 850 °C.

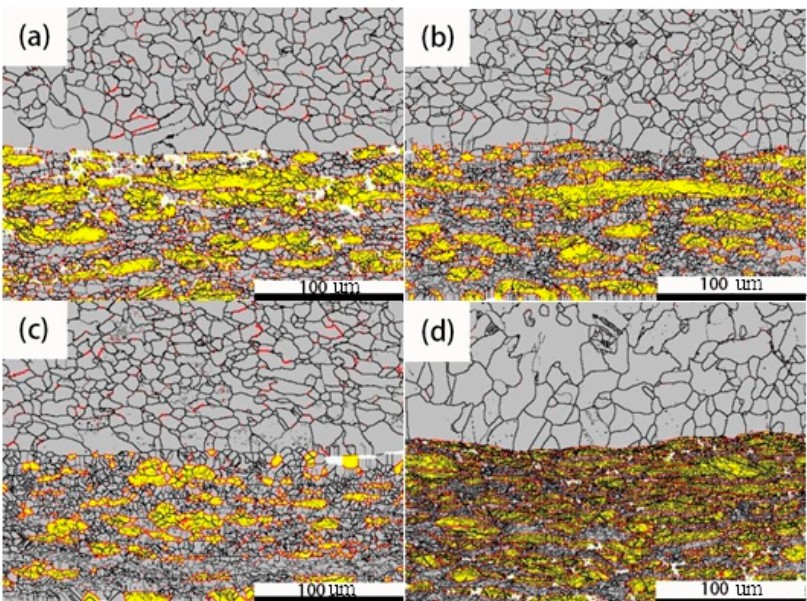

**Figure 7.** Phase boundary and grain boundary diagrams of large deformation regions at different rates at 850 °C. (**a**) 0.01 s$^{-1}$; (**b**) 0.1 s$^{-1}$; (**c**) 1 s$^{-1}$; (**d**) 10 s$^{-1}$.

In the figure, gray marks the bcc structure, yellow stands for the austenite fcc structure, the red grain boundary represents the phase boundary, the black coarse grain boundary represents >15°, and the black fine grain boundary represents 2–15°. First, the upper layer Q345 is basically composed of a bcc structure. The phase boundary first decreases and then increases with the increase of the deformation rate at a strain rate of 0.01–1 s$^{-1}$. The phase boundary is more obvious and uniform at 1 s$^{-1}$. When the strain rate increases to 10 s$^{-1}$ and the size of Q345 grain becomes larger, the analysis may be because the crystal is too late for dynamic recovery and recrystallization at high speed, resulting in coarse grains. By observing the lower layer 2205, it is discovered that at a strain rate of 0.01 s$^{-1}$, the fcc austenite phase presents an elliptical shape. As the deformation rate increases to 0.1 s$^{-1}$, the austenite phase shows a large block morphology and the local fine red phase boundary reduction. When the deformation rate increases to 1 s$^{-1}$, the large austenite phase is squashed, indicating a tendency of fragmentation. At this time, the proportion of bcc and fcc is coordinated. When the deformation rate increases to 10 s$^{-1}$, the austenite phase is severely elongated and deformed, and the morphology fragmentation becomes more obvious. From Figure 7, it is obvious that at 850 °C, as the strain rate increases, the black marked sub-grain boundaries (2–15°) gradually increase, from a local increase to the entire deformation zone.

Figure 8 exhibits the different morphological characteristics of the phase and grain boundaries at the interface of the 2205/Q345 thermal compression composite sample under different strain rates at 1000 °C.

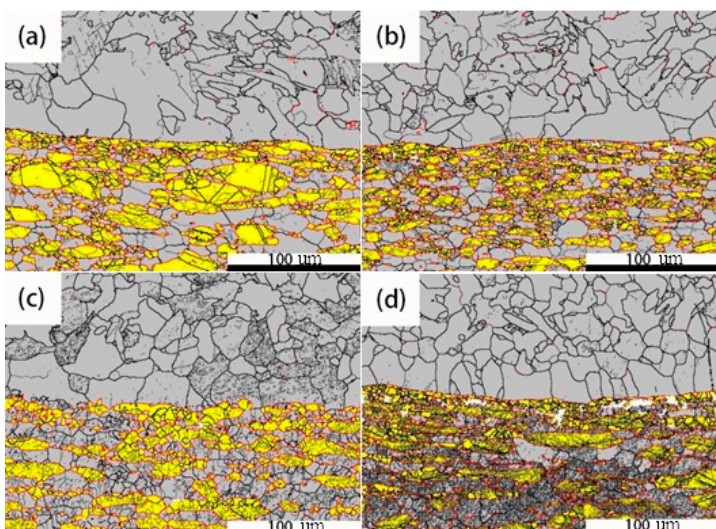

**Figure 8.** Phase boundary and grain boundary diagrams of large deformation regions at different rates at 1000 °C. (**a**) 0.01 s$^{-1}$; (**b**) 0.1 s$^{-1}$; (**c**) 1 s$^{-1}$; (**d**) 10 s$^{-1}$.

In the figure, gray is the bcc structure, yellow means the austenite fcc structure, and the red grain boundary is the phase boundary, the black coarse grain boundary is >15°, and the black fine grain boundary is 2–15°. First of all, the upper layer Q345 is composed of bcc ferrite. At a strain rate of 0.01 to 1 s$^{-1}$, the grain size decreases as the deformation rate increases. By observing the lower layer 2205, it can be seen that with the increase of deformation rate in the range of 0.01–1 s$^{-1}$, the austenite phase is continuously squashed from the bulk morphology, which shows a tendency of fragmentation. The proportion of bcc and fcc is relatively consistent at 1 s$^{-1}$. When the deformation rate increases to 10 s$^{-1}$, the austenite phase is severely elongated and deformed, and the morphology fragmentation becomes more obvious.

Figure 9 displays the different morphological characteristics of the two phases at different strain rates at 1050 °C.

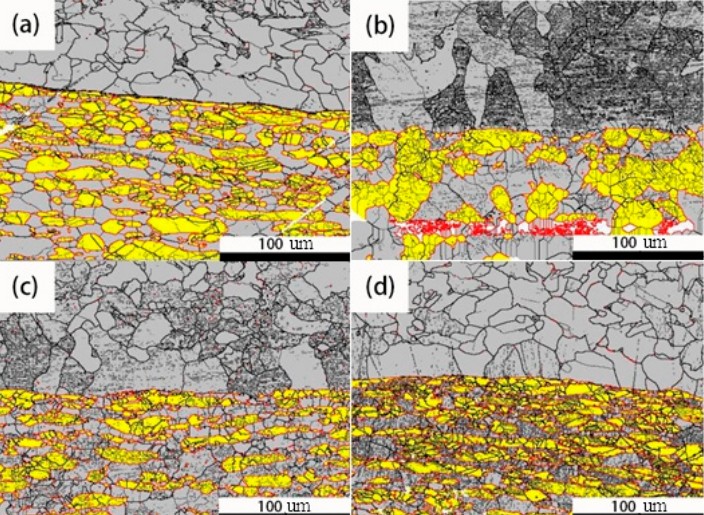

**Figure 9.** Phase boundary and grain boundary diagrams of large deformation regions at different rates at 1050 °C. (**a**) 0.01 s$^{-1}$; (**b**) 0.1 s$^{-1}$; (**c**) 1 s$^{-1}$; (**d**) 10 s$^{-1}$.

It is found from the figure that at a strain rate of $0.01\ \text{s}^{-1}$, the austenite phase has been crushed and is elliptical, and the sub-grain boundaries in the austenite fcc phase and the ferrite bcc phase are not obvious. However, with the increase of the deformation rate to $0.1\ \text{s}^{-1}$, the austenite phase is not uniformly distributed, and the degree of deformation is not large. At $1\ \text{s}^{-1}$, for 2205 the austenite fcc phase and the ferrite bcc phase are distributed in harmony, while when it increases to $10\ \text{s}^{-1}$, the austenite phase increases significantly, and its distribution range is greatly larger than that of ferrite.

### 4. Conclusions

In this paper, the isometric compression test is used to study the hot compression composite hot deformation behavior of 2205 duplex stainless steel/Q345 carbon steel, and the evolution of the structure is as follows:

(1)   There is a difference in deformation resistance between 2205 duplex stainless steel and Q345 carbon steel. Q345 carbon steel has a small deformation resistance. Therefore, in the process of deformation together, the carbon steel deforms first, with large degrees of deformation, recrystallization, and driving force. The grain is isometric and relatively large. The 2205 dual-phase steel was subsequently deformed, the degree of deformation was relatively small, and the microstructure retains the original rolled and elongated structure. At the composite interface, under the action of shear stress, there are a lot of fine grains.

(2)   At a low temperature of 850 °C, Q345 grains continue to be refined as the deformation rate increases. At 1000 °C, with the increase of deformation rate, the change of Q345 grains is not obvious. Therefore, 2205 and Q345 display a coordinated deformation trend as a whole at 1050 °C and $1$–$10\ \text{s}^{-1}$.

**Author Contributions:** Conceptualization, X.W.; analysis, X.W.; investigation, G.Z.; data curation, P.L.; writing—original draft preparation, X.W.; writing—review and editing, X.W. and G.Z.; supervision, J.L. and C.Z.; funding acquisition, P.L., G.Z., J.L. and C.Z. All authors have read and agreed to the published version of the manuscript.

**Funding:** This project was supported by the Innovation and entrepreneurship training program for college students of Taiyuan University of Science and Technology (XJ2021070), the Scientific and Technological Innovation Programs of Higher Education Institutions in Shanxi (2021L292), Fundamental Research Program of Shanxi Province (20210302124009 and 20210302123207), the Science and Technology Major Project of Shanxi Province (20181101017) and Taiyuan University of Science and Technology Scientific Research Initial funding (20212073 and 20212072). The authors would like to thank the Provincial Special Fund for Coordinative Innovation Center of Taiyuan Heavy Machinery Equipment for providing the facilities for the experimental works.

**Institutional Review Board Statement:** Not applicable.

**Informed Consent Statement:** Not applicable.

**Data Availability Statement:** Not applicable.

**Conflicts of Interest:** The authors declare no conflict of interest.

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
