# Peer review of "Thermal Deformation Behavior and Interface Microstructure Analysis of 2205/Q345 Hot Compression Composite"

_crystals, doi:10.3390/cryst12020301_

Round 1
Reviewer 1 Report
Dear authors Your paper is I general good but before publication have to be corrected. First of all please explain why were chosen cylindrical samples? From what I understand You are simulating flat products. If so using Gleeble 3800 system better effects are obtained for flat products the Hydrawedge modulus. Please explain this. Ad figure 3 do authors have observed any cracks on samples borders? Point 3.2 For the aim of this paper this part is not necessary. Since authors do not present any numerical simulation this part do not give anything to the paper and do not match to the title. Figure 7 is impossible to correctly analysis. The samples view ae too small and too dark. Since point 3.2 is not needed the conclusions have to be rewrite. Ad references, please look for literature not only from China. Problems of bimetallic materials are well known and are widely described. Add references from other part of the world.Author Response
Thanks for the reviewer giving me the opportunity to revise. Based on your comment and request, we have made extensive modification on the original manuscript.
- The thermal simulation compression experiment on Gleeble 3800 is a basic experiment. It is a unidirectional compression test. There are some differences between it and the actual situation.
- Figure 3has been modified. There is no crack in the compressed sample, which is the deformation behavior of the two metals at different temperatures and rates.
- Point 3.2 has been remove. The conclusions have beenmodified.
- Figure 5(the original is figure 7) has been modified.
- Referencehas been added foreign literatures and the introduction has been modified.

Reviewer 2 Report
In this paper in order to explore the thermal cladding process of 2205 duplex stainless steel/Q345 carbon steel, adopted thermal simulation method to systematically study the high temperature thermal deformation behavior of 2205 duplex stainless steel and Q345 carbon steel that was carried out under different temperatures and strain rates.
In principle, the manuscript is correctly written.
However, there seems to be a need to make a few corrections before publication.
There are a number of typing errors in the article, for example Authors start a sentence with a lowercase letter. This is unacceptable and needs improvement.
In Figure 3, the graphics quality is very poor. Actually, it is not known what this drawing is for, the more that the description of
"Thermal Compression Composite Test"
is incomprehensible.
The data presented in Figures Fig.4 and Fig.5 and Fig.6, are also poorly legible, the more so that there are no units in the descriptions of the X-Y axis.
In the formulas (1), (2), (3), what does the thick dot mean?
This is incomprehensible in the notations used in mathematical equations.
It can be published after corrections have been made.
Author Response
Thanks for the reviewer giving me the opportunity to revise. Based on your comment and request, we have made extensive modification on the original manuscript.
Point 1: There are a number of typing errors in the article, for example Authors start a sentence with a lowercase letter. This is unacceptable and needs improvement.
Response 1: The typing errors in the article have been modified.
Point 2: .In Figure 3, the graphics quality is very poor. Actually, it is not known what this drawing is for, the more that the description of Thermal Compression Composite Test" is incomprehensible.
Response 2: Figure 3 has been modified. The description has been changed to “2205 / Q345 thermal compression composite macro sample”.
Point 3: The data presented in Figures Fig.4 and Fig.5 and Fig.6, are also poorly legible, the more so that there are no units in the descriptions of the X-Y axis. The data presented in Figures Fig.4 and Fig.5 and Fig.6, are also poorly legible, the more so that there are no units in the descriptions of the X-Y axis.
Response 3: Point 3.2 for the aim of this paper is not necessary and not match to the title. Further, point 3.2 has been remove.

Reviewer 3 Report
The abstract is not very general, it does not clearly introduce the topic and purpose of the article, it only conculdes the results.
Please check english language and typesetting.
Is the sample only placed on top of each other or fixed somehow as a true compound?
Regarding water cooling to retain form: what about arising eigenstrains and phase transformation during cooling? Does this effect the final shape of the specimen?
In Fig. 3, the quality should be improved. Same viewing angle should be usewhen displaying the structure. Conclusion of similarity only with bare eyes or measurements?
All variables in all equations must be introduced, missing in e.g. Eq. 1-3
The point in Sect. 3.2 does not become clear. What is the goal here?
Section 3.3: What does OM microstructure mean?
Author Response
Thanks for the reviewer giving me the opportunity to revise. Based on your comment and request, we have made extensive modification on the original manuscript.
Point 1: The abstract is not very general, it does not clearly introduce the topic and purpose of the article, it only conculdes the results.
Response 1: The abstract has been modified. The high temperature thermal deformation behavior of 2205 / Q345 specimen at 850-1100 ℃ and strain rate of 0.01-10s-1 was systematically studied by the Gleeble-3800 thermal simulator, which provided a theoretical basis for the optimization of 2205 duplex stainless steel composite plate. It is found that the deformation resistance of 2205 steel was different from that of Q345 steel. Therefore, the Q345 steel deformed first, and the degree of deformation was large, the degree of recrystallization occurred, and the grain was isometric and relatively large. The 2205 steel was subsequently deformed, the degree of deformation was relatively small, and the microstructure retained the original rolled and elongated structure. In particular, 2205 and Q345 show a coordinated deformation trend as a whole at 1050 ℃ and 1-10s-1. Under the action of shear stress, there are many fine grains at the composite interface.
Point 2: Please check english language and typesetting.
Response 2: Paper has been checked and typeset.
Point 3: Is the sample only placed on top of each other or fixed somehow as a true compound?
Response 3: The two samples were placed on top of each other. Then samples subjected to bidirectional compression.
Point 4: Regarding water cooling to retain form: what about arising eigenstrains and phase transformation during cooling? Does this effect the final shape of the specimen?
Response 4: Quenching after thermal deformation has no effect on the final shape. This paper mainly analyzes the macro and micro changes at the interface.
Point 5: In Fig. 3, the quality should be improved. Same viewing angle should be usewhen displaying the structure. Conclusion of similarity only with bare eyes or measurements?
Response 5: Fig. 3 has been modified. The main purpose of Figure 3 is to show whether the two materials are consistent in the deformation process and what is its relationship with temperature and rate.
Point 6: All variables in all equations must be introduced, missing in e.g. Eq. 1-3. The point in Sect. 3.2 does not become clear. What is the goal here?
Response 6: Point 3.2 For the aim of this paper this part is not necessary and not match to the title. Further, point 3.2 has been remove.
Point 7: Section 3.3: What does OM microstructure mean?
Response 7: Section 3.3 has been modified to “ Microstructure of composite interface in large deformation area by optical microscope”

Round 2
Reviewer 1 Report
The paper can be published